# The Role of Pea (*Pisum sativum*) Seeds in Transmission of Entero-Aggregative *Escherichia coli* to Growing Plants

**DOI:** 10.3390/microorganisms8091271

**Published:** 2020-08-21

**Authors:** Leonard S. van Overbeek, Carin Lombaers-van der Plas, Patricia van der Zouwen

**Affiliations:** Wageningen Plant Research, Wageningen University and Research (WUR), 6700 AB Wageningen, The Netherlands; carin.lombaers@wur.nl (C.L.-v.d.P.); patricia.vanderzouwen@wur.nl (P.v.d.Z.)

**Keywords:** entero-aggregative *Escherichia coli*, *Pisum sativum*, EHEC, seed, plant production system, human pathogen, *E. coli* O104:H4

## Abstract

Crop plants can become contaminated with human pathogenic bacteria in agro-production systems. Some of the transmission routes of human pathogens to growing plants are well explored such as water, manure and soil, whereas others are less explored such as seeds. Fenugreek seeds contaminated with the entero-hemorrhagic *Escherichia coli* O104:H4 were suspected to be the principle vectors for transmission of the pathogen to sprouts at the food-borne disease outbreak in Hamburg and surrounding area in 2011. In this study we raised the questions of whether cells of the entero-aggregative *E. coli* O104:H4 strain 55989 is capable of colonizing developing plants from seeds and if it would be possible that, via plant internalization, these cells can reach the developing embryonic tissue of the next generation of seeds. To address these questions, we followed the fate of strain 55989 and of two other *E. coli* strains from artificially contaminated seeds to growing plants, and from developing flower tissue to mature seeds upon proximate introductions to the plant reproductive organs. *Escherichia coli* strains differing in origin, adherence properties to epithelial cells, and virulence profile were used in our experimentation to relate eventual differences in seed and plant colonization to typical *E. coli* properties. Experiments were conducted under realistic growth circumstances in greenhouse and open field settings. Entero-aggregative *E. coli* strain 55989 and the two other *E. coli* strains were able to colonize the root compartment of pea plants from inoculated seeds. In roots and rhizosphere soil, the strains could persist until the senescent stage of plant growth, when seeds had ripened. Colonization of the above-soil parts was only temporary at the start of plant growth for all three *E. coli* strains and, therefore, the conclusion was drawn that translocation of *E. coli* cells via the vascular tissue of the stems to developing pea seeds seems unlikely under circumstances realistic for agricultural practices. Proximate introductions of cells of *E. coli* strains to developing flowers also did not result in internal seed contamination, indicating that internal seed contamination with *E. coli* is an unlikely event. The fact that all three *E. coli* strains showed stronger preference for the root-soil zones of growing pea plants than for the above soil plant compartments, in spite of their differences in clinical behaviour and origin, indicate that *E. coli* in general will colonize root compartments of crop plants in production systems.

## 1. Introduction

Contamination of crop plants by human pathogenic bacteria can already occur at plant growth in the field or greenhouse. The transmission routes via which these pathogens are transmitted to crop plants are not always clear, but contamination can occur via (fresh) manure application to soil, upon irrigation, or via human handling with contaminated materials, equipment or machines. Alternative transmission routes of human pathogens to plants are less explored and one of these can be via starting materials such as seeds.

Seeds used for production of sprouts were reported to be contaminated with human pathogens such as *Escherichia coli*, especially the enterohemorrhagic variants thereof (EHEC), *Salmonella enterica, Bacillus cereus, Listeria monocytogenes, Staphylococcus aureus* and *Yersinia enterolitica* [1]. Human pathogens of different strains of *S. enterica*, *E. coli* O157:H7 and *E. coli* O104:H4 were shown to attach to fenugreek, alfalfa, lettuce and tomato seeds within 5 h upon exposure [2]. Furthermore, it was shown that *S. enterica* and *E. coli* O157:H7 strains could persist on butterhead lettuce seeds for two years and after that time, both pathogens were still able to proliferate on germinating seeds [3]. From these cases it is clear that seeds can become contaminated with human pathogens and that contaminated seeds can pose serious risks for human health at primary production of freshly consumed vegetables and fruits.

The largest outbreak related to consumption of produce was in Hamburg and the surrounding area (Germany) in 2011 upon consumption of fenugreek sprouts. Most likely, it was the fenugreek seeds used for sprout production that were the source of the contaminating agent, *E. coli* O104:H4, although the pathogen could never be traced back in the seed batches that were used for sprout production [4]. However, the outbreak strain was exceptional as it did not belong to the more ‘classical’ EHEC outbreak strains such as *E. coli* O157:H7, but instead to the group of entero-aggregative *E. coli* [5]. Commonly, *E. coli* O104:H4 has a human reservoir and is endemic in central Africa, but has never been reported before to be carried over via plants to humans. Entero-aggregative *E. coli* strains are dissimilar from the more common EHEC serotypes that carry the *eae* gene on a plasmid of which the product, the outer membrane protein intimin, is responsible for causing the typical intimate attachment to epithelial cells. *Escherichia coli* O104:H4 do not produce the intimin protein, but instead form a typical stacked brick-like cell structure on the intestinal cells and from there the bacterial cells release Shiga toxin into the human gut [6]. The aggregative adherence fimbriae (AAF), expressed from a gene located on the pAA plasmid, are responsible for this typical stacked brick bacterial cell feature, enabling entero-aggregative *E. coli* strains to colonize the intestinal mucosa layer. It is a question that still needs to be addressed whether these differences in bacterial attachment to human epithelial cells might also be responsible for eventual different behaviour of *E. coli* strains on seeds and on, or inside, plants emerging from seeds.

Plants can be colonized by *E. coli* and persistence near plants, on their outside surface and even internally, by different *E. coli* strains has been reported over time [7,8,9]. *Escherichia coli* strains show chemotactic responses to plants and can resist local plant immune responses in the form of oxidative stress [10,11] and it is able to protect itself against osmotic stress by conversion of plant derived choline to betaine [12]. *Escherichia coli* strains were found at different locations in multiple plant species and most commonly *E. coli* showed preference for the root zones and rhizosphere soils [13,14,15]. However, soil management also played an important role in persistence of *E. coli* in agricultural systems as the type of soil amendment on *E. coli* persistence was stronger than presence or absence of maize roots [16]. Damaging of seminal roots and root hairs, either mechanically, or caused by the phytopathogenic root-knot nematode *Meloidogyne hapla,* did not result in significant higher root internalization by *E. coli*, with respect to the undamaged control treatment [17]. Most of the described studies in plants were undertaken with *E. coli* O157:H7 and occasionally with *E. coli* K12, but not so often with strains belonging to other *E. coli* types, such as entero-aggregative *E. coli* strains to which *E. coli* O104:H4 belong to.

*Escherichia coli* is a complex species group and in the past this group was classified on the basis of their colonization behaviour in warm-blooded animals and on the clinical symptoms derived from oral ingestion of pathogenic strains. However, the *E. coli* species complex consists of strains whose lifestyles diverge from pathogens to commensals, and their capacity to persist and colonize ecological niches outside the colon is remarkable [18]. *Escherichia coli* genomes are flexible, allowing strains easily to adapt to local reigning circumstances [19,20,21]. Therefore, it should be accounted for that *E. coli* strains present in soil ecosystems are distinct from the ones in animals [22,23]. Human pathogens taxonomically related to *E. coli*, such as *Salmonella enterica*, also can colonize plants and even can communicate with plant cells [7,24,25]. Plants must be recognized as ‘alternative’ habitats for food-borne enteric bacteria [26] and Enterobacteriaceae, the taxonomical family to which *E. coli* and *S. enterica* belong to, are common inhabitants of plants. Particular groups of endophytes and phytopathogens also belong to this group [21,27] and therefore Enterobacteriaceae must be considered as an important component of the plant microbiome. Microbiomes of plants will form an important barrier against invasions of human pathogens in plant production systems and Enterobacteriaceae also play an important role in the defence of plants against invasive micro-organisms [28,29,30]. Human pathogens invasive to plants may acquire genetic information from plant-indigenous Enterobacteriaceae making them better adapted to circumstances locally reigning in, and near plants [21,31].

In this study we raise the questions as to whether *E. coli* O104:H4 is capable to colonize developing plants from seeds and if it would be possible that, via plant internalization, *E. coli* cells could reach the embryonic tissue of developing seeds. If the last is possible, then it would mean that, in the worst case, *E. coli* O104:H4 could persist in seeds for over many generations and even could be protected from common seed surface disinfection treatments because of the protection provided by the seed coat. Application to plant reproductive organs with the endophyte strain *Paraburkholderia phytofirmans* PsJN lead to internal seed contamination resulting in dissemination of this strain to next generations of plants [32]. Via experimentation, we explored the possibility if this could also be the case for *E. coli*. To address this question, we followed the fate of strain 55989 and of two other *E. coli* strains from artificially contaminated seeds to growing plants, and from developing flower tissue to mature seeds upon proximate introductions to the plant reproductive organs. We chose for Afila type (dwarf) pea, *Pisum sativum*, as model plant because of its fast and robust growth under variable circumstances, but also because of the size and smooth surface structure of the seeds making them easy to handle under the applied experimental circumstances. Experiments were conducted under realistic growth circumstances in greenhouse and open field settings. *Escherichia coli* strains differing in origin (plant versus human), adherence properties to epithelial cells (aggregative adherence fimbriae versus surface protein intimin), and virulence profile (possession of EHEC virulence genes or not) were used in our experimentation to relate eventual differences in seed and plant colonization to typical *E. coli* properties.

## 2. Materials and Methods

### 2.1. Escherichia coli Strains

Entero-aggregative *E. coli* O104:H4 strain 55989 was isolated in the period between 1996 and 1999 from faeces of a patient suffering from human immunodeficiency virus (HIV) and persistent diarrhoea from the Central African Republic [33]. This strain was used in all experiments conducted in this study. Two other strains were used for comparison in pea plant colonization: extended spectrum beta lactamase *E. coli* strain 0611 and Shiga toxin producing *E. coli* O150:H2 strain N112 (Table 1). Strain 0611 was obtained from morning glory leaves imported as culinary herb from Thailand [34], whereas strain N112 was obtained from grass plants taken from a meadow land located in the neighbourhood of Nijkerk, the Netherlands [23]. All *E. coli* strains were grown overnight under shaking (180 RPM) at 37 °C in Luria Bertani broth (LB: Tryptone, 10 g; yeast extract, 5 g; NaCl, 10 g; dissolved in one L water and standard autoclaved for 20 min at 121 °C). Thus obtained late exponential cells were harvested by centrifugation and cell pellets were washed twice in Ringer solution (¼ strength Ringers, OXOID BR0052, Basingstoke, UK; one tablet dissolved in 0.5 L demineralized water followed by autoclaving). Prior to experimentation, optical densities at 600 nm (OD_600_) of suspensions were measured and retrospectively validated by colony forming unit (CFU) enumeration onto Brilliance *E.coli*/coliform selective agar (BECSA, OXOID CM1046, Basingstoke, UK) medium without further amendments for strain 55989, or onto BECSA amended with 1 µg per ml of cefotaxime for strain 0611 or onto CHROMagar^TM^ O157 (CHROMagar^TM^, Paris, France) for strain N112.

### 2.2. Pea Seed Treatment with E. coli

The same seed batch of Afila type (dwarf) pea (*Pisum sativum*) were used throughout all experiments. Microbial analysis of seeds revealed a contaminating fungal load in the seed lot and, therefore, seeds were disinfected with 1% chlorine solution for 10 min. As a result of disinfection, the germination rate of the seeds increased from 84.8% to 93.2%. Disinfected seeds were, therefore, used in all further experiments.

For setting up a standardized seed inoculation protocol, the minimal exposure time and cell density required for efficient attachment of *E. coli* cells to the seed surface, performed with strain 55989 only, were established. For establishment of the minimal exposure time, seeds were exposed for 1, 5, 10, 30 and 60 min to exponentially-grown Log_10_ (Log) 5.59 *E. coli* strain 55989 cells per mL and upon exposure, treated seeds were stored for 24 h at room temperature, after which seeds were rinsed twice in sterile Ringer solution to remove loosely attached strain 55989 cells from the seed surface. For establishment of the minimal effective density of strain 55989 cells for seed treatment, seeds were exposed for one h to a range of inocula, 10-fold increasing in densities, starting from Log 0.37 to up to Log 7.37 cells per mL, using sterile water as negative control. After exposure, treated seeds were stored for 24 h at room temperature, after which seeds were washed two times in sterile Ringers solution and 100 µL of washing solutions and smashed pea seed extracts (made by hammering one seed for 30 s in a Bioreba bag [Bioreba AG, Reinach, Switzerland] with three mL Ringer solution) were plated onto BECSA, and plates were incubated under standard conditions, i.e., for 16 h at 37 °C for colony formation. For inactivation of strain 55989 cells attaching to the pea seed surface, three different disinfection solutions; i.e., silver/copper (AMS), chlorine dioxide and hypochlorite, as previously described for lettuce leaf decontamination [35], were applied. Therefore, 10 µL droplets containing Log 5.79 strain 55989 CFUs were placed onto seed surfaces and incubated for one h at room temperature. Upon incubation, droplets were removed with a pipet tip and treated seeds (*n* = 5 per treatment) were incubated for one min in the different disinfecting solutions using sterile water as control. Upon incubation, liquids and smashed pea extracts were plated onto BECSA, and plates were incubated.

For experimentation in greenhouse and fields, two types of seed inoculation methods were applied: (1) local seed inoculation using 10 µL droplets with *E. coli* cells (drop inoculation) placed on the seed surface and (2) submergence of seeds in an *E. coli* cell suspension (seed submergence). For drop inoculation, a single droplet with between Log 7.2 and 7.4 cells per 10 µL were placed on the seed surface and treated seeds were kept in a laminar flow cabinet at room temperature for one h, after which droplets were removed using a sterile pipet tip. For submergence, seeds were incubated for one h in a laminar flow cabinet at room temperature in a washed *E. coli* cell suspension with densities between Log 9.2 and 9.4 cells per mL. Seeds from all treatments were collected, washed twice in Ringer solution and dried overnight (16 h) in a laminar flow cabinet. For the second field study, conducted in 2016 and executed with strain 55989 only, submerged seeds were washed, dried and stored in a desiccator for 48 h at constant relative air humidity (RAH) of 36% or 96%, or stored for 35 d in a fridge at 4 °C, at ambient RAH. Relative air humidities in the desiccators was maintained at 36% by using a saturated CaCl_2_ solution and at 96% by using demineralized water.

### 2.3. Comparison of Three Different E. coli Strains in Pea Plant Colonization

A plant colonization study with three *E. coli* strains (55989, 0611, N112) in developing pea plants was performed in the greenhouse. For that purpose, seeds were treated with the three strains separately or with water (control) via submergence and drop inoculation (*n* = 5 per treatment, three strains plus control and two inoculation types, leading to a total of eight treatments). A total of 160 treated, and 80 non-treated (see later) seeds were planted into standard potting soil and treated seeds were allowed to germinate in the dark for three d followed by subsequent plant growth at a day/night regime of 8 h darkness at 18 °C and 16 h light at 22 °C with a constant air humidity of 70%. Densities of the different *E. coli* strains, using water-treated seeds as controls, were determined by plating onto the agar medium respective for each strain and on BECSA for control seeds, for eventual presence of indigenous *E. coli* in seeds. Based on different growth stages after 8, 19, 34, 50, 62 and 80 d, plants of all treatments were analysed at different locations for the eventual presence of (inoculated) *E. coli*. On day 8, the primary roots, cotyledon and epicotyl were analysed, whereas at days 19, 34 and 62 roots, stem base (stem 0.5 cm above the soil until the first true leaf pair) and top stem (stem part between the fourth and fifth true leaf pair) were analysed. At 80 days after sowing (DAS) seeds were collected for later analysis on eventual presence of *E. coli*. Primary roots, cotyledons and epicotyls at day 8 were taken from plants and transferred to Bioreba bags containing one mL Ringer solution and extracts were made by hammering as described before for treated seeds. These extracts were then serial diluted 10-fold, and non-diluted and diluted samples were plated onto the agar medium respective for each strain and on BECSA for extracts from control plants, and all plates were incubated under standard conditions. Root and stem (stem base and top) samples, taken at 19, 34 and 62 days after sowing (DAS), were transferred to Bioreba bags with three mL Ringer solution after which extracts were made. Non-diluted and tenfold serial diluted samples were then plated onto the different agar media after which *E. coli* colonies were allowed to grow out.

Non-treated plants (80 in total) at day 45 were inoculated with each of the three *E. coli* strains and water (as control) in the flower receptacle, or in the first leaf axil below the developing flower receptacle (*n* = 5, three strains plus control and two inoculation types leading to eight treatments) in separate plants to investigate whether local inoculation with *E. coli* could lead to (internal) seed infection. Therefore, wounds were created by punching with a sterile needle into the flower receptacle and leaf axil tissues, and five µL droplets, containing no (control) or between Log 6.4 and 6.6 *E. coli* CFUs, were placed on top of these wounds. Two hours after applications, the droplets were taken up by the plants and five and 35 d after inoculation, at 50 and 80 DAS respectively, treated plants were analysed for the local presence of *E. coli* at the inoculation sites and/or in developing pods and seeds. At 50 DAS, only developing pods were analysed for plants inoculated at flower receptacles, whereas for plants inoculated at the leaf axils, developing pods, leaf axil and internode tissue were sampled. At 80 DAS, developed pods and seeds were sampled for both treatments. Samples taken from all tissues, of between 0.3 and 2 g in weight, were further processed and plated on agar media for *E. coli* colony formation. Ripened seeds from plants of all 16 treatments, taken at 80 DAS (one seed per plant; *n* = 5 per treatment) were collected and stored at 4 °C for later analysis on presence of *E. coli*.

### 2.4. Colonization of Pea Plants by Strain 55989, Introduced onto Seeds, under Open Field Circumstances

Field studies in two consecutive years were conducted with pea plants grown from water (control) and strain 55989-treated seeds in the periods between 28 May and 20 August 2015 and between 26 May and 17 August 2016. Both studies were performed in an agricultural field at the Unifarm experimental farm of Wageningen University and Research centre (GPS coordinates, 51°59′18.0″ N; 5°39′40.1″ E). The field was 4 m × 6 m in size and was covered by a net to avoid entrance of birds into the field. The soil at the Unifarm field location was a sandy soil consisting of 86% sand, 9% silt, 1% clay, organic matter content of 3.9% and pH of 5.9. In the first field experiment, one seed treatment was applied; i.e., seeds submerged in strain 55989 solution. Seeds submerged in water served as control and for each sampling, five plants per treatment and sample time were used. Plants were sampled for analysis in the presence of strain 55989 in rhizosphere soil, roots, stem base and seeds. For the second field experiment, five seed treatments were applied that are realistic under commercial seed treatment and storage circumstances: (1) submergence, (2) drop inoculation, (3) drop inoculation followed by 48 h storage at 96% RAH, (4) same as 3, but then stored at 36% RAH, and (5) submerged and stored for 35 d at 4 °C. Seeds submerged in water were used as control, leading to six treatments using five plants per treatment and sampling time. At 0 DAS (28 May 2016) seeds were sown into the soil and at 28, 49, 62 and 83 DAS, plants were sampled and analysed for presence of strain 55989 in rhizosphere soil, roots and seeds. Sampling times were chosen on the basis of the same plant developmental stages as in the first field experiment. At the first, second, third and fourth samplings in 2015 and 2016, plants were, respectively, in the seventh true leaf stage, in the flowering stage, at the end of flowering/beginning of the pod filling stage, and at the senescent stage with ripened seeds. Plant development was slower in the second than in the first year of field experimentation, due to lower temperatures in 2016, and therefore samples were taken at later time points after sowing in 2016 than in 2015. For each sampling, plants with roots were taken from the soil and soil loosely attached to roots was removed by shaking. Intact plants were separately packed in plastic bags and transported to the laboratory where all samples were processed on the same day. Soil firmly attached to roots was considered to be ‘rhizosphere soil’ and per plant between one and seven g mixed roots with soil were shaken in sterile 50 mL plastic tubes filled with one g gravel and 20 mL 0.1% sodium pyrophosphate solution. Tubes were vortexed for one min and soil suspensions were either plated non-diluted onto BECSA, or in 10-fold serial dilutions made in 0.1% sodium pyrophosphate solution. Then, roots were removed from the tubes, two times rinsed in sterile demineralized water and pat dried on sterile filter paper. Stem base parts (only taken in 2015 and between 0.2 g and maximally 3 g in weight) and rinsed roots (between 0.5 and 4 g) were transferred to Bioreba bags containing three mL Ringer solution, after which stem base and root parts were hammered and thus derived extracts were plated undiluted and tenfold serial diluted. Individual purple stained colonies, indicative for strain 55989, at the last samplings in 2015 and 2016 (respectively, at 85 and 83 DAS) were streaked to purity onto BECSA and single colonies were grown overnight in LB at 37 °C. Resulting dense cultures were either mixed with sterile glycerol to a final concentration of 20% for storage at –70 °C, or cells from one mL culture suspensions were pelleted by centrifugation after which DNA was extracted from resulting cell pellets for whole genome sequencing, using the Illumina MiSeq platform. Comparison of obtained isolates with strain 55989 and other *E. coli* strains, as described in van Overbeek et al. [23], was made on the basis of DNA sequences of 10 cellular household genes, i.e., adenylate kinase (*adk*), classII fumarate hydratase (*fum*C), glycerol kinase (*glp*K), DNA gyrase subunit B (*gyr*B), 3-isopropylmalate dehydrogenase (*icd*), diaminopimelate decarboxylase (*lys*A), malate/lactate/ureidoglycolate dehydrogenase (*mdh*), methionine-tRNA ligase (*met*G), adeylosuccinate synthetase (*pur*A), and DNA recombination/repair protein (*rec*A).

Seeds, five per treatment (two in field experiment 1 and six in field experiment 2) from different plants taken at the last samplings in both years, were collected and stored at 4 °C for later analysis for eventual presence of *E. coli*.

### 2.5. Analysis for Presence of E. coli in Surface-Sterilized Seeds

All collected seeds from greenhouse and both field experiments, coming to a total of 120 seeds, were surface disinfected using 1% chlorine solution as described before. Final wash water was plated onto the agar medium respective for each *E. coli* strain, using BECSA for water-treated controls. Disinfected seeds were allowed to germinate on sterile water-soaked filter paper in Petri dishes for 6 d in the dark at room temperature. Then, emerging plants were transferred to Bioreba bags containing 3 mL Ringers solution, hammered and resulting extracts were directly plated onto the agar medium respective for each strain, using BECSA for extracts from plantlets grown from water-treated seeds (control extracts). In parallel, these control extracts were also spiked with strain 55989 (to an estimated final density of Log 4 CFU per mL) and plated onto BECSA as a control for the establishment of eventual *E. coli* growth inhibition in pea seed extracts.

### 2.6. Data Processing and Statistics

*E. coli* densities on seeds, in root, pod or stem tissue, and in rhizosphere soil were expressed as Log_10_ (*n* + 1) values per seed, per g of plant or per g of dry soil, respectively. Prevalence was defined as the number of samples with detectable (introduced) *E. coli* CFUs. When absent in a sample, introduced *E. coli* was considered to be undetectable and the lowest limit of detection was calculated on the basis of lower than one single colony on duplicate plates that had received non-diluted extracts. These numbers are expressed as ‘0’ values. Geometric means were calculated from Log-converted values and used for statistical analysis of variance (ANOVA, Genstat 19th Ed. Hemel Hempstead, UK). Differences were considered to be different at levels of *p* = 0.05 and below.

## 3. Results

### 3.1. Attachment of E. coli Strain 55989 Cells to Pea Seeds

Strain 55989 cells attached to pea seeds within one min upon exposure. Of the Log 5.59 CFUs per seed used for exposure, on average Log 2.52 CFUs firmly attached to the seeds, even after two washes with Ringer solution, which is a fraction of 0.09% of the total inoculum added to seeds (Appendix A). Elongation of the exposure time to five, 10, 30 and 60 min resulted in higher average values of, respectively, Log 2.66, 3.69, 3.29 and 3.63 CFU per seed, and the average cell number attached after one minute on seeds was significantly lower than after 10 and 60 min, but not after five and 30 min. The highest fraction (1.2% of the inoculum) of strain 55989 cells attaching to pea seeds was observed at 10 min and after that time the fraction of attaching cells remained about stable. A linear relationship existed between the density of cells applied to seeds for 60 min and the ones still attaching to seeds after two washes (Figure 1). Based on these data, it was decided to choose for the highest possible inoculum densities, by making use of late exponential *E. coli* cell cultures consisting of Log 9 CFU per mL and over, upon a standard exposure time of 60 min. Two seed inoculation procedures were applied throughout this study, i.e., seed submergence and spot inoculation and the main differences between the two inoculation types are that spot inoculation resulted in a local contamination on the seed surface, which may better reflect contaminations occurring under practical circumstances, and that the inoculum dose was about Log 1.2 CFU per seed lower for spot than for submergence inoculation. Strain 55989 cells, attached to pea seeds, were still sensitive to three different disinfection reagents (Appendix A). This indicates that the strain 55989 cells attached to seeds were accessible for these reagents. Strain 55989 cells on dried pea seeds persisted for 35 d at 4 °C (a temperature realistic for seed storage in practice) on the seed surface at about the same level (on average Log 6.83 CFU per seed) as at the moment of introduction onto the seeds (Log 6.56 CFU per seed). Also, strain 55989 cells attached to seeds could persist with low air humidity (36% RAH) for 48 h. When compared to inoculated seeds stored for the same time period at 96% RAH, the number of strain 55989 cells on seeds was slightly (Log 0.3 CFU per seed), but not significantly lower. In conclusion, application of strain 55989 cells on pea seeds resulted in a robust and persistent contamination of the seed surface.

### 3.2. Colonization of Pea Plants Emerging from Seeds by Three E. coli Strains

Log CFU numbers per seed at the time of sowing (0 DAS) were 8.66, 8.64 and 8.47 on seeds submerged in suspensions of, respectively, strains 55989, 0611 and N112, whereas upon drop inoculation numbers were, respectively, 7.34, 7.31 and 7.39. Eight days after sowing, *E. coli* CFUs were recovered from the cotyledon although numbers were between Log 0.43 and 3.38 per seed lower than at 0 DAS (expressed in Δ CFU values, see Figure 2). Overall, *E. coli* CFUs per cotyledon from the seed submergence treatments (6.22) were higher than from the drop inoculation treatments (3.22), whereas no significant differences were observed between the different strains within each seed inoculation method. Primary roots at 8 DAS were already colonized by the three strains, irrespective of the applied inoculation method, although on average higher *E. coli* Log CFU numbers per g plant were on roots developed from ‘submerged’ seeds (4.55) than on the ones developed from ‘drop inoculated’ seeds (3.14) (Figure 3). No significant differences on primary roots were observed between strains for each seed treatment method. Over time, from 19, 34 and 62 DAS, both *E. coli* prevalence (i.e., *E. coli* was present at a detectable level in the sample) and cell density levels (expressed in geometric averages over the five samples) on primary and secondary roots equally declined for all three *E. coli* strains. Overall, prevalence and Log CFU numbers per g root were higher for plants grown from ‘submerged’ than from ‘drop inoculated’ seeds, but still at 62 DAS, colonies from all three strains were recoverable from roots from both seed inoculation methods. This in contrast to *E. coli* CFUs from epicotyls at 8 DAS, where Log CFU numbers per g plant were between 0 (undetectable) and 5.06 (Figure 3). There were clear distinctions in *E. coli* prevalence and cell density levels in epicotyls between plants grown from ‘submerged’ and from ‘drop inoculated’ seeds; i.e., *E. coli* was found in 10 of 15 plants at an average density of Log 1.94 CFU per g plant in epicotyls developed from submerged seeds, whereas in 2 of 15 plants at average density of Log 0.56 CFU per g plant in epicotyls developed from ‘drop inoculated’ seeds. Over time from 19 to 62 DAS, *E. coli* prevalence and cell density numbers in stem base samples declined, although CFUs from all three strains were recoverable in stem base samples taken at 62 DAS from plants grown from the submerged seeds only. Over the same time period, no *E. coli* CFUs were found in the top stem samples indicating that none of the applied *E. coli* strains systemically spread to higher located places in the pea stems. Therefore, it is unlikely that it can be assumed that *E. coli* cells applied to seeds would lead to systemic spread throughout the entire plant finally resulting in invasion of plant reproductive organs. Therefore, it was investigated whether local introductions of *E. coli* strains at the start of flower development could lead to (internal) seed contamination.

Introduction of *E. coli* cells to receptacles of developing flowers and to the first leaf axil below developing flowers in untreated plants at 45 DAS led to consistent high recovery of CFUs of all three *E. coli* strains at the inoculation sites at 50 DAS (Figure 4). However, further systemic spread from the leaf axil to internode tissue at 50 DAS was limited as *E. coli* prevalence (between 1–3 out of 5 replicate plants per strain) and density levels (between Log 0 and 3.90 CFU per g plant) were low. Thirty five days after flower receptacle treatment, *E. coli* CFUs for all three strains were almost consistently found in pods at between Log 0 (one plant treated with strain N112) and 6.44 CFU per g plant. However, no *E. coli* CFUs were found in pods from plants inoculated at the leaf axils. Furthermore, no *E. coli* CFUs were found in water, collected from all treatments (strain x seed inoculation method), after the last washing after seed disinfection and in plantlets emerging from disinfected seeds, indicating that no internal seed contamination had taken place. However, in control samples that were spiked with strain 55989 cells, *E. coli* CFUs were found on plates upon incubation, demonstrating that the absence in *E. coli* colony formation was not the result of *E. coli* growth inhibition by compounds present in plant extracts. Therefore, it must be concluded that local plant contamination, even to most proximate places of the plant reproductive organs, did not lead to internal seed infection by any of the three investigated *E. coli* strains.

### 3.3. Colonization of Pea Plants by E. coli Grown from Seeds Treated with Strain 55989 Cells under Open Field Circumstances

Experiments over two successive years were conducted with pea seeds treated with strain 55989 and (water-treated) control seeds in the open field in soil under fluctuating temperature, water and air humidity regimes, in the omission of any treatment for plant pest and disease control. Under these circumstances, pea plant development was comparable as under greenhouse circumstances, although plants were between 10–15 cm smaller in the open field than in the greenhouse. In the first field experiment, performed in 2015, seeds were treated with strain 55989 cells by submergence, resulting in a load of Log 8.79 CFU per seed at 0 DAS. In control plants, strain 55989 CFUs were never found at any time point during the experiment in root, stem, rhizosphere soil and seed samples. After 18 DAS, strain 55989 CFUs were recovered from roots from 4 out of 5 plants grown from strain 55989 seeds at an average density of Log 2.33 CFU per g plant (Figure 5). Later at plant growth, strain 55989 prevalence and densities in roots declined, and only one sample (of 8) was found positive for strain 55989 at 85 DAS, when plants were at a senescent stage. Only one stem sample (of five) was found to be positive for strain 55989 at 18 DAS, at a level of Log 1.54 per g plant, in a plant grown from a strain 55989-treated seed. No strain 55989 CFUs were recovered from stem samples at later plant growth stages. This to the contrary to rhizosphere soil, where strain 55989 CFUs were more consistently present. At 18 DAS, strain 55989 density level was on average Log 3.14 CFU per g dry soil, but this density declined at later plant growth stages and at 85 DAS one, of eight, investigated plants was found positive for strain 55989. Strain 55989 thus had a stronger preference for colonization of the root zone than for the above-ground plant compartments. Genomic DNA of two purified isolates taken from roots and rhizosphere soil at 85 DAS revealed that both isolates were identical to strain 55989 on the basis of 10 *E. coli* household genes. However, one other isolate from rhizosphere soil sampled at 85 DAS, with a colony morphology slightly deviating from strain 55989, was genotypically not identical to strain 55989, but clustered with other *E. coli* strains in the same phylogenetic tree. The presence of strain 55989 in roots and rhizosphere soil of senescent pea plants was therefore confirmed, although also another *E. coli* strain was present in rhizosphere soil that was not applied in the seed treatment. No *E. coli* CFUs were found in plantlets emerged from ripened (dry) and surface disinfected seeds collected from strain 55989 plants.

The second field experiment, conducted in 2016, was focused on root zone and seed colonization by strain 55989 only, using five different strain 55989 and one control treatments. Seed loads with strain 55989 at 0 DAS were (in Log CFU per seed): 8.24, 7.36, 8.36, 8.04 and 6.83, respectively, for the following seed treatments: (1) submerged, (2) drop inoculated, (3) submerged and 48 h stored at 96% RAH, (4) submerged and 48 h stored at 36% RAH and (5) submerged and stored for 35 d at 4 °C. Over all sampling points, no strain 55989 CFUs were found in rhizosphere soil, roots and seeds of control plants. In roots sampled at 28 DAS, highest prevalence was found in plants grown from drop-inoculated seeds and seeds stored for 48 h at 96% RAH (presence of strain 55989 in all five plants for both treatments; Figure 6). For the other three treatments at 28 DAS, strain 55989 was present in only one or two of the five analyzed samples (Figure 6). Prevalence (between zero and four) and density values (between Log 0 and 2.02 strain 55989 CFU per g plant) dropped over time in roots at 49 and 62 DAS and at 83 DAS only one sample was found positive at a density level of Log strain 2.51 strain 55989 CFU per g plant (seed treatment at 36% RAH for 48 h). Similar to the first field experiment, strain 55989 was also found present in rhizosphere soil. At 28 DAS, strain 55989 CFUs were found in almost all rhizosphere soil samples, with the exception of one from plants grown from a seed stored for 35 d at 4 °C (Figure 6). Density levels in rhizosphere soil at 28 DAS were significantly higher when plants were grown from submerged seeds and seeds stored for 48 h at 96% RAH (at respectively Log 2.30 and 2.51 strain 55989 CFU per g dry soil) than in the ones grown from seeds that were stored for 35 d at 4 °C, or stored for 48 h at 36% RAH, or that were drop inoculated (respectively, at levels of Log 1.42, 1.67 and 1.98 CFU per g dry soil). Over time, prevalence of strain 55989 dropped, occasionally to below three positive samples per treatment. However, the high prevalence at 62 DAS in roots of plants grown from the seeds stored for 48 h at 36% and at 96% RAH (respectively, five and four of five tested samples were found positive for strain 55989), indicated that dry storage of contaminated seeds does not affect plant colonization by *E. coli* strain 55989. At 83 DAS, prevalence was lowest of all sampling points and strain 55989 CFUs were only recovered from two samples; one from plants grown from drop-inoculated seeds, and the other from seeds stored at 36% RAH for 48 h. Of the seven isolates from rhizosphere soil (two) and roots (five) at 83 DAS, four (two from rhizosphere soil and two from roots) were identical to strain 55989 on the basis of gene sequences of 10 *E. coli* cellular household genes. The other three isolates (all from roots) were different from strain 55989, but still were identified as *E. coli*. Plantlets emerged from collected and surface-sterilized seeds from all treatments and taken at last samplings at 83 DAS (five seeds from different plants per treatment, six treatments, 30 seeds in total) in the second field experiments were all negative, confirming the observation made in the greenhouse experiment and the first field experiment that transmission of strain 55989 cells via the mother plant to internal compartments of the next generation of seeds must be considered as an unlikely event.

## 4. Discussion

Entero-aggregative *E. coli* strain 55989 was able to colonize the root compartment of pea plants from inoculated seeds. In the root and rhizosphere soil, the strain could persist until the senescent stage of plant growth, when seeds had ripened. Colonization of the above-soil parts of the plants by strain 55989 was negligible and, therefore, translocation of strain 55989 cells via the vascular tissue of the stems to developing seeds seems unlikely. The fact that even proximate introductions of cells of all three strains to developing flowers did not result in effective internal seed contamination led to the conclusion that there is no evidence for internal transmission of *E. coli* via plants to developing seeds. The consequence thereof is that *E. coli* O104:H4 will not be transmitted as internal seed contaminant to next generations of plants, at least not in pea plants. However, as external seed contaminant, *E. coli* O104:H4 will be transmitted to next generations of plants and its ecological behaviour in plants is the same as for the two other *E. coli* strains. This is an important aspect because it demonstrates that pathogenic *E. coli* strains can contaminate crop production sites by making use of seeds as primary vectors for transmission to growing plants. The soil environment plays an important role in further transmission of these pathogens to edible parts of the plants. Our finding corroborate that of Habteselassie et al., [14], where it was demonstrated that the rhizosphere played an important role in the early establishment and colonization of radish and lettuce plants by *E. coli* O157:H7.

The two *E. coli* phylotype B1 strains used in our study (O104:H4 strain 55989 and O150:H2 strain N112) were indistinguishable in their ecological behaviour in, and near developing pea plants, this in spite of their differences in pathotype. The behaviour of the two phylotype B1 strains on their turn were indistinguishable from the third one, the ESBL-producing *E. coli* strain 0611 that originated from plants. Niche partition and adaptation to natural habitats has been hold responsible for diversification of different *E. coli* clades in their ecological behaviour in many natural habitats such as in soils [18,19,22,36], but most likely this does not hold true for plant ecosystems. In plants and their surrounding areas (rhizosphere soil), plant-released nutrients are available for cell proliferation and like many bacteria naturally occurring in soils, *E. coli* profit from available nutrients explaining the long-term persistence near pea roots and in rhizosphere soils as demonstrated in both field experiments. The three *E. coli* strains declined over time in the pea rhizosphere which contrasted the ecological behaviour of the typical soil-borne bacterial species *Pseudomonas fluorescens* that persisted at constant and higher levels in the pea rhizosphere [37]. Although not a species fully adapted to soil environments, *E. coli* responded here in an opportunistic fashion to nutrients made available by plants, either at seed emergence or at root exudation. The fact that *E. coli* isolates were found in pea rhizosphere soil and roots that differed from the introduced strain 55989 indicate that *E. coli* is intrinsically present in pea plants or their proximate surroundings. Therefore, these strains must be considered as indigenous inhabitants of the pea plant microbiome. This would also indicate that these *E. coli* strains are adapted to life near or even inside plants. The presence of *E. coli* in plants is still an unknown fact and would require further attention in eventual later studies. Altogether, this emphasizes the fact that plants must be considered as alternative ecosystems for *E. coli* [26].

The above soil compartments of the pea plants were less colonized than the roots by all three *E. coli* strains, indicating that these parts of the plants are less favoured for colonization by *E. coli* in general. Most likely nutrient availability was limited at the epicotyl from where later at plant growth the stem base developed. Possibly a part of the *E. coli* populations colonized the internal tissue of the stem base, but because no surface sterilization attempts were undertaken in our study, no information can be provided on exact numbers of *E. coli* cells internalized into plants. Internalization of spinach, lettuce and *Nicotiana benthamiana* by *E. coli* O157:H7 was reported in Wright et al. [9] and internalization was indicated to play a critical role with respect to food safety and human health [8]. However, food safety was not the principal motive for our research, but instead it was the ecological behaviour in, or on plants, in relation to colonization of developing seeds. From our study we can conclude that either via external, or internal colonization, even upon introduction shortly before flower formation, no internal seed colonization by any of the three studied *E. coli* strains took place. From there, it can be concluded that *E. coli*, under realistic growth circumstances, is not able to cross the barrier between maternal and embryonic tissue in pea plants. This in contrast to a typical endophyte, *Paraburkholderia phytofirmans* PsJN, that was able to colonize internal seed compartments of maize, soy and pepper upon flower spraying [32]. Most likely, *E. coli* lack the appropriate cellular equipment to independently penetrate plant tissue like endophytes do. Penetration of *E. coli* into seeds can be forced, e.g., by vacuum infiltration [38], but these are circumstances that do not occur under realistic plant growth circumstances.

From seeds, *E. coli* and *S. enterica* strains were shown to colonize cotyledon and the primary roots of emerging alfalfa, fenugreek, lettuce and tomato plants [38], in a fashion comparable to what was shown in our study with the three *E. coli* strains in pea plants. In the study done by Liu and co-workers [38], the pathogens were present under the seed coat, whereas in our study, the *E. coli* strains were applied on the seed coat. It seems that no difference exist in colonization of plants either by *E. coli* pathogens present inside or on the surface of seeds. In our study, the time needed for attachment to pea seeds was within one minute upon exposure and no differences in efficiency of attachment to the pea seed surface was observed between the three strains. In Cui et al. [2], it was reported that differences in seed attachment between *E. coli* O157:H7 and O104:H4 was present on different seed types (alfalfa, fenugreek, lettuce and tomato). *Escherichia coli* O104:H4 strain ATCC BAA2326 (Hamburg 2011 outbreak strain) showed the lowest attachment potential in comparison with three *E. coli* O157:H7 strains. Lowest attachment may be related to relative higher EPS production by *E. coli* O104:H4 strains, favouring biofilm formation, but restricting adherence to solid surfaces [2]. Also, higher cell hydrophobicity and rougher (more wrinkled) seed surface type often result in higher attachment of bacterial pathogens to seeds [2]. So, in spite of the facts that: (i) *E. coli* has a relatively lower surface hydrophobicity, (ii) *E. coli* O104:H4 is less efficient in attachment to solid surfaces, and (iii) that the surface of pea seeds is rather smooth, we still observed long-term persistence to up to 35 days of *E. coli* strain 55989 on the pea seed surface, and this strain was also shown to stay alive at low RAH of 35% for 48 h on the pea seed surface. In line with observations made by van der Linden et al. [3], it must be concluded that *E. coli* can persist over long periods in time on seed surfaces and these time periods and storage conditions are realistic in commercial seed production and logistics circumstances. It can therefore be concluded that presence of *E. coli* on seeds, used for starting materials in fruit and vegetable production, pose a potential risk for food safety unless these seeds are appropriately disinfected.

It was demonstrated before that pea seeds can become contaminated with human pathogens as demonstrated at the *Campylobacter jejuni* outbreak in Alaska in 2008 [39]. In case human pathogens are present on seeds used as starting materials for cultivation of crop plants, then a risk on contamination of the edible parts of the plants may exist. From contaminated seeds, human pathogens can reach rhizosphere soil from where they can contaminate edible parts grown below the soil surface, as will be the case for radish and carrot plants, or the edible parts shortly above the soil surface, as will be the case for leafy greens. With regard to the consequences on long-term persistence of pathogenic *E. coli* strains in soils of plant production systems, nothing is certain at the moment. Phytopathogens may create ports of entrances for human pathogens from where these pathogens further spread to the edible parts. However, no internalization occurred in spinach plants by *E. coli* O157:H7 when two pathogens, *Pseudomonas syringae* and *Meloidogyne hapla,* were present in the production site [17], indicating that plant pathogens do not play a dominant role in contamination of plants by human pathogenic *E. coli*. However, gene fluxes in *E. coli* are high [18,36] and auxiliary genomes of *E. coli* are often prone to homologous recombination [20]. Adaptation of pathogenic *E. coli* strains to the plant environment might result from gene acquisition events taking place in plants and their surrounding areas [18,31]. On the longer run this may result in selection of new types of pathogenic *E. coli* strains with improved capacities to colonize plants [21,31]. Therefore, longer persistence of human pathogens in plant environments might become a concern for fresh food production.

In summary, no evidence was provided on seed internalization via growing plants by the entero-aggregative *E. coli* strain 55989 and the other two tested *E. coli* strains. Internalization of *E. coli* into seeds, followed by transmission to next generations of crop plants, must be regarded as an unlikely event under realistic plant growth circumstances. However, *E. coli* strain 55989 could persist on the pea seed surface and was shown to survive under commercially applied seed storage conditions. All three *E. coli* strains showed stronger preference for the root-soil zones of growing pea plants than for the above soil plant compartments, in spite of their differences in clinical behaviour and origin. Therefore, these observations can be expanded to broader groups of (pathogenic) *E. coli* strains with the concept that contaminating *E. coli* strains will colonize root compartments of crop plants in production systems and from there are able to further spread to consumable parts of plants. Adequate seed disinfection protocols should avoid contamination of arable soils used for cultivation of freshly consumable crop plants.

## Figures and Tables

**Figure 1 microorganisms-08-01271-f001:**
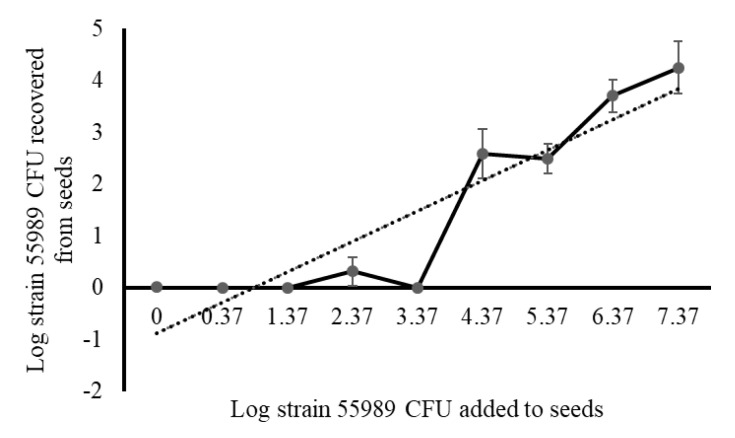
Relationship between the density of *E. coli* strain 55989 cells exposed to pea seeds and number of cells attaching to the seed surface. Seeds were exposed to different strain 55989 cell densities for 60 min, where after treated seeds were two times washed prior to analysis for presence of strain 55989 cells that remained attached to the seed surface. Dashed line represents the linear regression line (y = 0.6219x − 0.6588; R^2^ = 0.7961).

**Figure 2 microorganisms-08-01271-f002:**
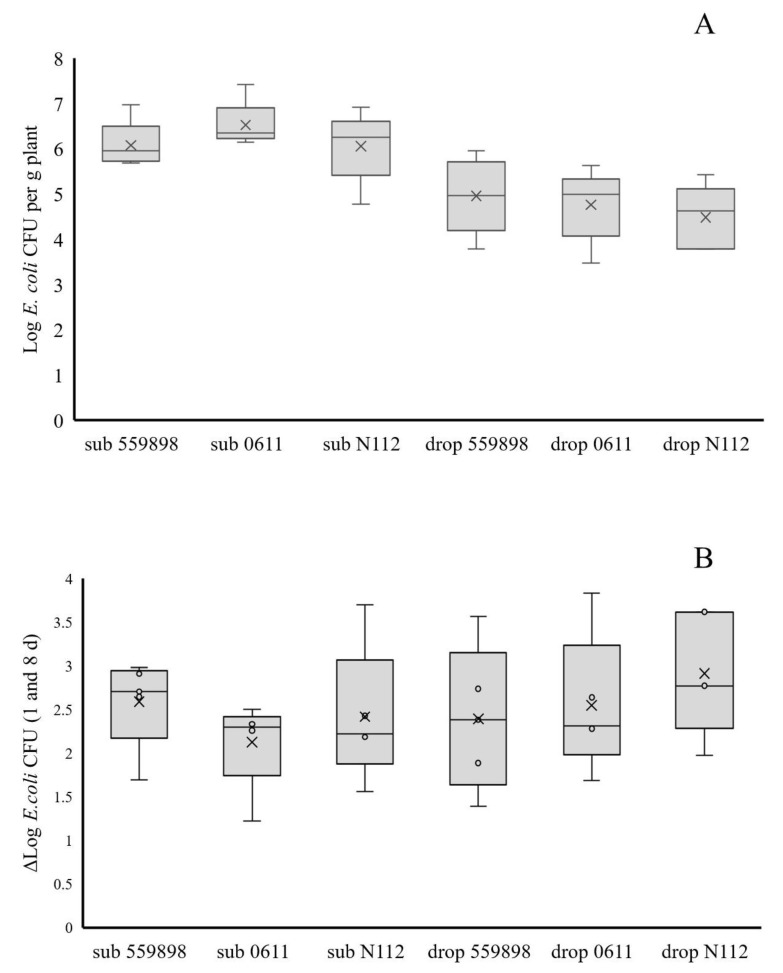
Persistence of three *E. coli* strains on pea cotyledons after 8 DAS (**A**) and the difference between *E. coli* colony forming units (CFUs) at 0 DAS on seeds and at eight DAS at the cotyledons, expressed as Δ Log *E. coli* CFU values (**B**). Cells of three *E. coli* strains (55989, 0611, N112) were administered to seeds via submergence (sub) or inoculation by placement of 10 µl droplets on the seed surface (drop).

**Figure 3 microorganisms-08-01271-f003:**
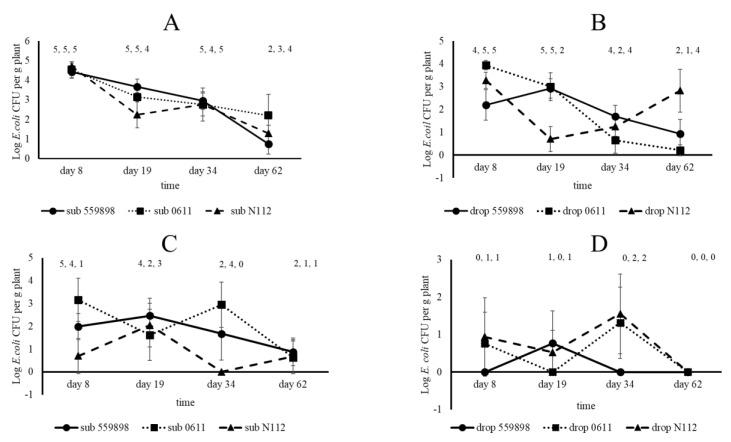
Colonization of roots (**A**,**B**) and epicotyl/stem base (**C**,**D**) of pea plants grown from seeds treated with *E. coli* strains 55989, 0611, or N112, either via seed submergence (**A**,**C**) or drop inoculation (**B**,**D**). Symbols: circle, strain 55989; square, strain 0611; triangle, strain N112. Numbers above symbols indicate the number of plants found to be positive for the presence of introduced *E. coli* strains, of five replicate plants.

**Figure 4 microorganisms-08-01271-f004:**
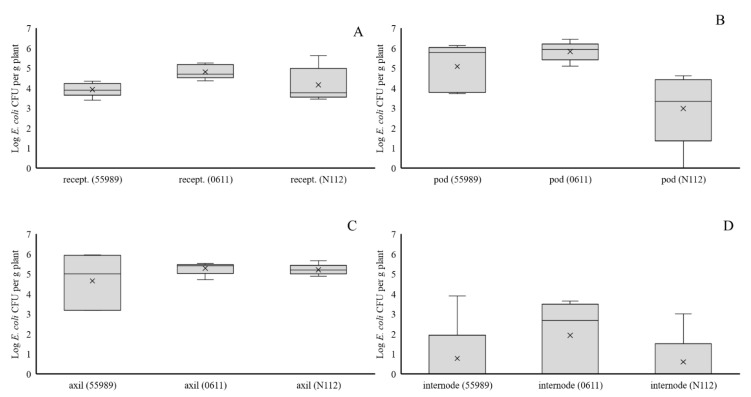
Colonization of flower receptacles at 50 DAS (**A**), pea pods at 80 DAS (**B**), leaf axils at 50 DAS (**C**) and first internode above the leaf axil at 50 DAS (**D**), upon injection of three *E. coli* strains (55989, 0611, or N112) into, respectively, flower receptacles (**A**,**B**) or leaf axils (**C**,**D**) at 45 DAS.

**Figure 5 microorganisms-08-01271-f005:**
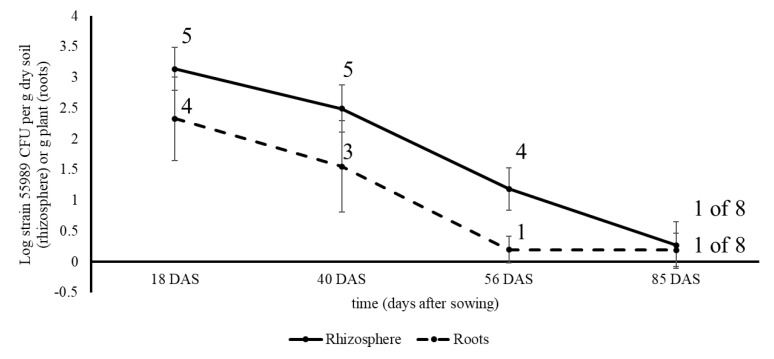
Colonization of roots and rhizosphere soil of pea plants by strain 55989, emerged from seeds treated with strain 55989 cells, under open field circumstances in 2015. Solid line, rhizosphere soil; dashed line, roots; numbers above symbols indicate the number of plants found positive for presence of strain 55989, of five replicate plants taken at 18, 40 and 56 DAS and of eight at 85 DAS.

**Figure 6 microorganisms-08-01271-f006:**
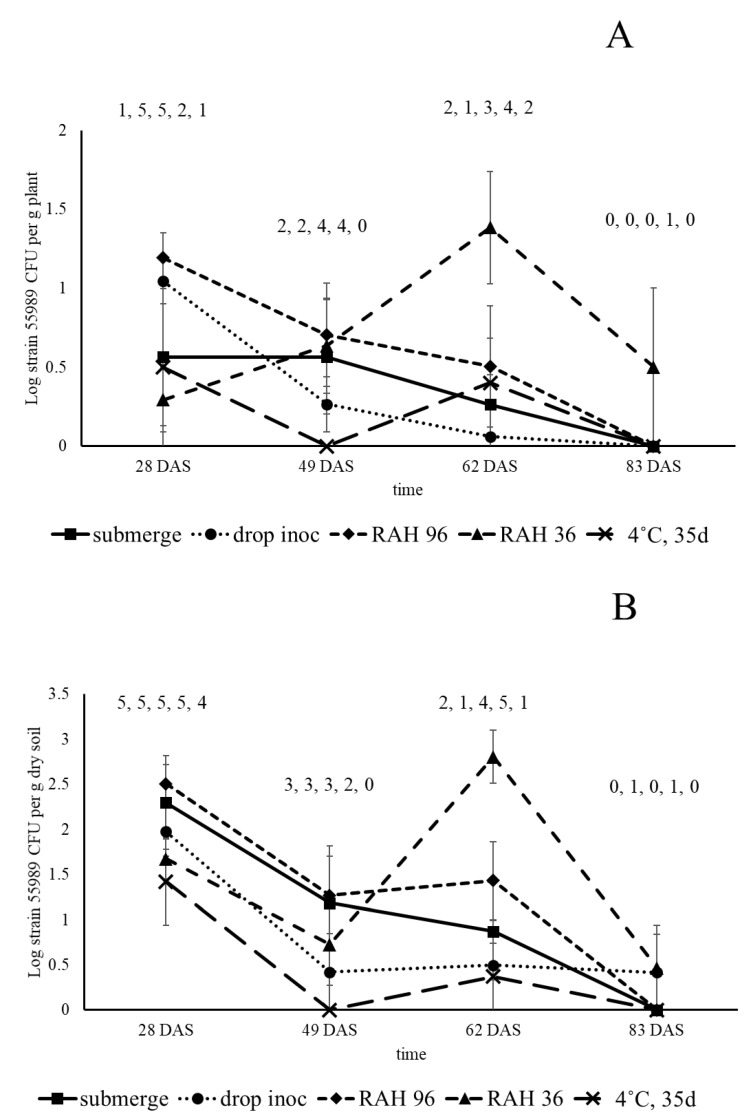
Colonization of roots (**A**) and rhizosphere soil (**B**) of pea plants by strain 55989, emerged from strain 55989-inoculated seeds, treated under different bacterial seed application and storage regimes, under open field circumstances in 2016. Square, seed submergence; circle, drop inoculation; diamond, seed submergence, followed by storage at 96% relative air humidity (RAH) for 48 h; triangle, seed submergence, followed by storage at 36% RAH for 48 h; cross, seed submergence followed by seed storage at 4 °C for 35 d. Numbers above symbols indicate the number of plants found positive for presence of strain 55989, of five replicate plants at all samplings.

**Table 1 microorganisms-08-01271-t001:** *E. coli* strains used in this study.

Strain	Serotype	Characteristics	Origin	Reference
55989	O104:H4	Entero-aggregative *E. coli*	human faeces	[33]
N112	O150:H2	STEC (*stx*1, 2, *hly*A, *eae*)	grass shoots	[23]
0611	not identified	ESBL (*chr*: *bla*_CTX-M-15_ *qnr*S1)	morning glory	[34]

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
