# Peer review of "The Role of Pea (Pisum sativum) Seeds in Transmission of Entero-Aggregative Escherichia coli to Growing Plants"

_microorganisms, 2020, doi:10.3390/microorganisms8091271_

Round 1
Reviewer 1 Report
This study aimed to determine whether an entero-aggregative E. coli O104:H4 (strain 55989) pea plants from seed, and whether the bacteria reached the next generation of seeds by internalisation of the plant tissue. They compared the colonisation & internalisation ability with two other E. coli strains to determine if pathotype affected the outcome.
This type of work is important because it helps to understand the colonisation dynamics for a priority foodborne pathogen, which in turn informs on risk assessment. It also adds valuable fundamental bioscience to the community working on the ecology of plant-microbe interactions, or on foodborne pathogens. The work is novel because they authors have been able to carry out field trials with a pathogenic bacteria (under appropriate conditions), which is the most relevant setting and thus adds value to the work.
Main Comments
This is a very useful and well conducted piece of work. The conclusions are sound and logically drawn from the findings, and it is well set in context. I have a few minor suggestions for improvement.
In general, the text is a little wordy and there is room for reduction. For example, the Introduction is quite long with some repetition, and contains extraneous information that whilst very interesting, is not directly relevant to the main question of the study (e.g. section L100 – 117). In the Results section, there are quite a few interpretative statements / sentences that better belong in the Discussion, rather than simply reporting on the findings (e.g. Lines 313, 320, 330), or additional information not required for the main question that can be condensed (e.g. L349-356; L581-597). Alternatively, simply combine the Results & Discussion sections.
I take a little uneasy with the statement (Line 524) ‘The ecological behaviour observed in pea plants with these three strains may therefore be extrapolated to the entire E. coli species complex’, because as the authors well know, the entire species complex is genetically vast with many divergent genotypes, which do exhibit different metabolic capabilities. This is borne out by the fact that the field-grown pea trials actually picked up endemic E. coli, while the level of the inoculated strain decreased (but this is entirely expected, where it would reach a low, but stable tick-over level once established in the native microbiome). I think we’d see a different outcome if for example the inoculated / test strains were at the extremes of the spectrum, e.g. UPEC or NMEC compared to rhizosphere-derived endemic E. coli isolate. Related to that (Line 534) please insert ‘…here, E. coli responded in an opportunistic fashion…’
Minor Comments / typos
Line 57: correct ‘were’ to ‘was’
Line 68: Change ‘It are’ to ‘it is’ or remove entirely
Line 83: Capitalise first letter of Meloidogyne hapla
Line 101: correctly ‘taxonomic’ to ‘taxonomically’
Line 139: I think it may be worth stating the year for this isolate, so that there is a clear understanding that it is not derived from the 2011 fenugreek-associated outbreak.
Line 146: change ‘Luria broth’ to Lysogeny broth’
Line 146: do you have an evidence that bacteria grown at 37 °C are affected in colonisation compared to growth at plant-relevant temperatures / room temperature?
Line 166: correct ‘10Log’ to ‘Log10’
Line 210: duplicated ‘for that purpose’
Line 211: define ‘DAS’. Presumably the same as DPI, days post infection / inoculation?
Lines 329,444: it is preferable to show the data rather than ‘data not shown’ for full transparency, even if in Supplementary files.
Line 349: rephrase for sense to ‘… roots and epicotyl had developed, the cotyledon was still intact, and growth of …’.
Line 467: rephrase to ‘Similar to the first field trial…’
Line 476: change to ‘Remarkably, …’, or better still remove and stick with ‘However, …’ / ‘In contrast’ / etc
Line 511: change ‘is playing’ to ‘plays’
Line 518: could simply state different pathotypes instead of more lengthy explaination?
Fig. 2B for clarity, please update the y axis to ‘ΔLog10CFU (d1/d8)’
Figs. 3 & 6 In general, with a low number of replicates, it is more useful to show individual values instead of means (meaningless for 2 or 1 reps). However, I appreciate it may become hard to visualise three separate tests (strains) as scatterplots, but please can you investigate if this is possible in alternative plots, e.g. as dotplots with connecting median lines?
Author Response
see attached Word document.

Reviewer 2 Report
This study tests for the possibility of E. coli of colonizing pea plants germinating from inoculated seeds, which could result in food-borne disease outbreaks as have been reported in other plants and bacteria. The results demonstrate limited colonization of above-ground plant parts making incorporation and transmission by seed unlikely, but there was more affinity for the bacteria in the soil-root zone, suggesting that root crops might pose a higher risk of contamination by E. coli.
Overall I found the paper to be clear and well-written but parts (methods especially) were very long (almost 200 lines of single-spaced text) and provided more details than needed on topics not strongly related to the focus of the paper (line 63-70, 90-99, 348-356), while not providing enough details on more relevant parts for this paper (102, 105-110, 245-248). In particular, the discussion of E. coli as a basic part of the plant microbiome needs more detail.
In many parts of the methods there were specific details and treatments related to time, temperature, humidity, media but it was not clear where these standards came from? Have they been optimized in previous studies, are just chosen? For example, 207-220, 251-259. The methods seem highly controlled and repeatable, but how were the particular parameters selected?
Regarding section 2.5, line 287 and beyond, what if pathogenic E. coli is epiphytic rather than endophytic, which seems to be the assumption. The surface sterilization of seeds and tissues could be washing away the bacteria that might still be widespread on the surfaces of field grown plants, and therefore potentially contribute to food-borne illness.
The paper generated some interesting results, such as the finding that (line 320) that inoculum densities were more important than time of exposure for adherence to seeds, and relative effectiveness of different inoculation methods (327-328), sensitivity to disinfection reagents (329) and differences among strains. The ability of the bacteria to colonize seeds, and to infect above and below ground tissues of developing plants and spread systemically within the plant help our understanding of this plant-bacterial interaction. However, the relevance for different inoculation methods and for different sites of inoculation for field-grown plants is not made clear. The data from field grown plants do indicate that the colonization of above-ground parts is limited compared to the root-soil interface. However, the authors don’t really address how pea seeds could become contaminated with E. coli to start with in an agricultural or garden setting. The discussion on 581-609 is very good and discusses the results of this study with larger issues of food-borne pathogens and human disease. In total, this paper advances our understanding of E. coli as a possible contaminant in plant-based agriculture.
Specific comments
Line 27 – temporal or temporary?
158 – does chlorine affect bacteria?
188 – seems like a short time. How were these particular parameters decided on?
Fig. S2 – needed?
Author Response
See attached Word document
